# Can We Predict Prostate Cancer Metastasis Based on Biomarkers? Where Are We Now?

**DOI:** 10.3390/ijms241512508

**Published:** 2023-08-07

**Authors:** Ignacio F. San Francisco, Pablo A. Rojas, Juan C. Bravo, Jorge Díaz, Luis Ebel, Sebastián Urrutia, Benjamín Prieto, Javier Cerda-Infante

**Affiliations:** 1Environ Innovation Laboratory, Avenida Providencia 1208 Oficina 207, Providencia, Santiago 7500000, Chile; benjamin@environ.bio; 2Servicio de Urología, Complejo Asistencial Dr. Sotero del Río, Santiago 8150215, Chile; projasr@clinicalascondes.cl; 3Servicio de Urología, Hospital Regional Libertador Bernardo O’Higgins, Rancagua 2820000, Chile; bravoi.jc@gmail.com; 4Servicio de Urología, Instituto Oncológico Fundación Arturo López Pérez, Santiago 7500921, Chile; jorge.diaz@falp.org; 5Servicio de Urología, Hospital Base de Valdivia, Universidad Austral, Valdivia 5090000, Chile; luisebel@uach.cl; 6Servicio de Urología, Hospital Dr. Hernán Henríquez Aravena, Universidad de La Frontera, Temuco 4780000, Chile; sebastian.iuv@gmail.com

**Keywords:** prostate cancer, metastasis, tissue-based genomic biomarkers

## Abstract

The incidence of prostate cancer (PC) has risen annually. PC mortality is explained by the metastatic disease (mPC). There is an intermediate scenario in which patients have non-mPC but have initiated a metastatic cascade through epithelial–mesenchymal transition. There is indeed a need for more and better tools to predict which patients will progress in the future to non-localized clinical disease or already have micrometastatic disease and, therefore, will clinically progress after primary treatment. Biomarkers for the prediction of mPC are still under development; there are few studies and not much evidence of their usefulness. This review is focused on tissue-based genomic biomarkers (TBGB) for the prediction of metastatic disease. We develop four main research questions that we attempt to answer according to the current evidence. Why is it important to predict metastatic disease? Which tests are available to predict metastatic disease? What impact should there be on clinical guidelines and clinical practice in predicting metastatic disease? What are the current prostate cancer treatments? The importance of predicting metastasis is fundamental given that, once metastasis is diagnosed, quality of life (QoL) and survival drop dramatically. There is still a need and space for more cost-effective TBGB tests that predict mPC disease.

## 1. Introduction

Worldwide, in 2020, an estimated 1,414,259 men were diagnosed with prostate cancer (PC) and 375,304 died from the disease at a crude rate of 9.5%. Approximately 15% of cases were metastatic at diagnosis, with a 5-year survival rate of 31% [1]. Once a diagnosis is made of non-metastatic, clinically significant PC according to risk criteria, a local curative treatment should be offered in order to reduce the risk of metastatic disease and eventually the risk of mortality [2,3,4]. On the other hand, if metastatic prostate cancer (mPC) is diagnosed, only palliative management can be used in order to improve survival and quality of life (QoL). Any curative intent is impossible at this stage, especially when patients are in the last stage of non-metastatic (nmCRPC) or metastatic castrate-resistant prostate cancer (mCRPC), despite multiple new drugs and tools that have been developed in the last few years [5,6,7,8,9,10,11,12]. Moreover, there is an intermediate scenario in which patients have no mPC but have initiated a metastatic cascade through epithelial–mesenchymal transition [13]. Nonetheless, even if we invested many resources, we would not be able to identify these patients. Current tools that are utilized with several limitations, such as clinical, biochemical and histological parameters, clearly are not sufficient to identify patients that will progress to mPC and those who will not [14,15]. In general, adjuvant treatments after local therapy using the above criteria have not shown better outcomes than observation or early salvage treatment [16,17]. From the perspective of the clinical setting, there are still several questions with no clear answers. For example, are local treatments such as radical prostatectomy (RP) or radiotherapy (RT), with or without androgen deprivation therapy (ADT), capable of killing or eradicating prostate cancer cells from the body? Should adjuvant therapy be used more frequently as a personalized therapy? Is prostate-specific membrane antigen (PSMA) positron emission tomography (PET)/computed tomography (CT) showing early metastatic disease clinically useful? Does it reduce mortality? Is there a better way to classify localized, high-risk patients? There is indeed a need for more and better tools to predict which patients will progress in the future to non-localized clinical disease or already have micrometastatic disease and, therefore, will clinically progress after primary treatment. This information could help to determine which patients may need multimodal or adjuvant treatment even with localized disease, and, in consequence, which patients do not need more than a single modality of treatment. It is well known that patients with unfavorable intermediate- and high-risk features are at a higher risk of recurrence and develop metastatic disease after primary treatment; therefore, this group of patients are likely the best candidates for the testing of new potential biomarkers to predict mPC.

Biomarkers are molecules that can provide information about the diagnosis, progression, prognosis and prediction of a pharmacological response. These include the presence of specific cell types, proteins, metabolites, RNA, DNA mutations, polymorphisms or epigenetic modifications [18]. Different types of biomarkers have been studied in PC as tools for the analysis of the risk of progression, with the potential for application in the accurate identification of candidates for active surveillance (AS), adjuvant therapy and/or new therapy modalities. We recently published a review of the role of biomarkers in AS [19]. This review included urine, blood and tissue biomarkers. In addition, several studies in the past few years have shown the role of imaging as a tool for the diagnosis, staging and therapy of mPC. In this vein, multiparametric magnetic resonance imaging (mpMRI) [20] and PET/CT [21,22,23] represent important tools in the diagnosis, staging and monitoring of PC patients.

The best approach for monitoring remains challenging for clinicians; biomarkers are being developed and may play an important role. Biomarkers for the prediction of mPC are still under development; there are few studies and not much evidence of their usefulness. This paper seeks to address the limited literature in the field.

We know that biochemical failure or persistent elevated prostate-specific antigen (PSA) after primary treatment and the appearance of clinical metastasis can take several years [24,25,26]. Currently, with the use of PSMA PET/CT, this time has been shortened [23]. However, PSMA PET/CT imaging is still under development and not universally available, and the evidence of its clinical utility, especially in reducing mortality, is still under research. Developing and better understanding the role of biomarkers in predicting metastatic disease may be essential in improving PC management and eventually reducing PC mortality. We could potentially obtain these biomarkers through needle biopsies or RP tissue specimens—years before the theorical appearance of metastasis—in order to cure patients before they develop metastasis, particularly those with a high risk of developing it or even with circulating tumor cells (CTC) waiting to nest in the bones and/or lymph nodes. To date, most of the literature related to tissue-based genomic biomarkers (TBGB) has been focused on Oncotype DX, Prolaris and Decipher. However, their role has been better demonstrated in AS rather than advanced disease [27,28,29,30,31,32]. This review is focused on TBGM for the prediction of metastatic disease. We develop four main research questions that we attempt to answer according to the current evidence. 

(i)Why is it important to predict metastatic disease?(ii)Which tests are available to predict metastatic disease? What is the evidence and the efficacy of commercialized tissue prognostic tests?(iii)What impact should there be on clinical guidelines and clinical practice in predicting metastatic disease?(iv)What are the current prostate cancer treatments and potential new therapeutic opportunities using TBGM for the prediction of metastasis?

## 2. Questions Development

### 2.1. Why Is It Important to Predict Metastatic Disease?

Metastatic hormone-sensitive PC (mHSPC) will inevitably lead to death in a 10-year period. Therefore, it is crucial to treat patients during localized disease and eventually before detecting any signs of clinical metastasis. Currently, biomarkers for the prediction of metastatic disease are being studied. Current tools are biochemical and histological variables such as PSA, the Gleason score, margin status and tumor volume [15]. In some cases, these variables are used to decide when to add adjuvant therapy as radiotherapy post-RP based on the tumor stage and margin status [33,34,35,36]. However, as with any treatment, the risk of overtreatment and the loss of QoL associated with the treatment are important elements to consider. Therefore, a biomarker needs to be accurate.

Much has been written about the mechanism of PC’s development into metastasis. Epithelial–mesenchymal transition [13] and circulating tumor cells are involved in this mechanism [37]. Bone metastasis is most frequently seen in PC because, in part, of the action of cytokines/chemokines and growth factors [38,39]. There is a latent period of many years between the metastatic cells leaving the prostate and the development of metastatic disease. The crucial time period in terms of curation is not precisely known. Is a cell that exits the prostate guaranteed to produce radiologic metastasis? What proportion of cases with CTC will develop metastasis? During what point in the disease will a biomarker have the best probability of being a good predictor and therefore allow the curing of a patient? 

mPC is by far the main cause of PC mortality. In the last few years, the incidence of mPC has increased from 18% to 25%. The age at diagnosis has decreased from 71 to 68 years of age, and non-Hispanic, Asian, low-income and unmarried men are more likely to die of mPC, respectively [40,41]. This is explained by less frequent PSA screening as a result of the recommendation of the US Preventive Service Task Force against routine PC screening for men of all ages in 2012 [42]. This point is relevant since the median survival of patients with a new diagnosis of mPC is 42 months with ADT alone. However, this population is heterogenous since the diagnosis can be provided as de novo oligometastases (defined as ≤3 to 5 metastases) or polimetastases, or metachronous metastatic disease after primary treatment for localized disease [9]. It has been found that men with prostate cancer metastases have a 29.8% five-year survival rate, as compared to a 100% survival rate in men with localized prostate cancer. The primary goal of any cancer treatment is to reduce the cancer-specific mortality. This is measured through metastasis-free survival, which is a surrogate endpoint for the risk of disease death [43] and overall survival [44].

Having mPC is not only related to mortality but also QoL. There is a long list of potential complications from metastatic disease, such as bone fractures, cord compression, lymph edema and renal failure secondary to urinary obstructions [45]. In addition, the treatment of androgen blockage production or action and/or radiotherapy can result in side effects that include fatigue, anemia, breast enlargement and tenderness, hot flashes, loss of libido, erectile dysfunction, loss of muscle mass and strength, fatigue, anemia, depression, hair loss, osteoporosis, fractures, obesity, insulin resistance, alterations in lipids and greater risks for diabetes and cardiovascular disease [46]. Other side effects are more specific, depending on which treatment is used—for example, abiraterone (hyperkalemia, elevation of liver enzymes), apalutamide (rash), enzalutamide (falls), docetaxel (febrile neutropenia) and PARP inhibitors (anemia and fatigue) [8,11,47,48,49]. Therefore, it is important to prevent and eventually detect and treat these adverse effects (AE)—for example, by preventing bone fractures with vitamin D/calcium, bisphosphonates or human monoclonal antibodies against RANKL. The effect on QoL has been well studied and demonstrated in this stage [50], and new therapies have demonstrated that they improve survival and also QoL [51]. Bone metastasis is by far the most frequent complication (80%) in PC [52]; therefore, bone health is a very important concept in men with metastases. Skeletal-related events (SREs) include pathologic fracture, spinal cord compression, palliative radiation or surgery to the bone and changes in antineoplastic therapy secondary to bone pain. All of these complications can dramatically affect QoL and survival. For example, a pathological bone fracture in prostate cancer patients increases mortality by 20% [53], and bone metastasis increases mortality by six times with no SRE and by 10 times with SRE [54]. Consequently, algorithms have been created to better treat and follow patients with ADT and/or bone metastasis in order to maintain bone health [55]. 

Strategies to better detect metastatic disease early on to minimize the AEs of systemic treatment and perhaps improve survival have been developed. PSMA PET/CT and directed metastasis therapy are good examples of these strategies, respectively.

**PSMA PET/CT versus conventional imaging.** PSMA PET/CT has been implemented in the past few years and it has shown better sensitivity, specificity and accuracy than conventional imaging (CT and bone scan) in the detection of metastatic disease [21,56]. However, it is still unclear if the improved performance of PET/CT will result in a greater cancer survival and overall survival rate. ^18^F-DCFPyL recently received FDA approval for the staging of biochemically recurrent prostate cancer. The main role of PSMA PET/CT has been shown in staging in high-risk patients [21] and also in detecting the location and number of metastases in men with biochemical recurrence after primary treatment with radical prostatectomy [23].

**Directed metastatic therapy (DMT).** One of the major concerns about treating patients with metastatic disease is the adverse effects of ADT, as described above. This is the main reason that DMT has been used for oligometastatic disease [57,58]. A series of studies have been published using PET and conventional imaging [22,57,58,59]. The objective is to delay the use of ADT, which probably does not affect the oncologic outcome. 

It is important to understand that predicting/preventing metastatic disease is crucial to survival because once a patient has metastatic disease, there is no curative window of time and treatment; there is only palliative treatment that can prolong life. 

### 2.2. Which Tests Are Available to Predict Metastatic Disease? What Is the Evidence and the Efficacy of Commercialized Tissue Prognostic Tests?

There are three TBGB available in the market that may play a role in predicting mPC, which will be discussed below.

**Oncotype DX** is a biopsy-based genomic test designed as a reverse transcription polymerase chain reaction assay that has been analytically validated to measure the expression of 17 genes in RNA extracted from fixed tumor tissue from prostate needle biopsies [32]. The test provides a Genomic Prostate Score (GPS) result, on a scale of 0–100, with increasing scores indicating more biologically aggressive disease. It has been clinically validated as a strong, independent predictor of adverse pathology (AP) (defined as Gleason score [GS] ≥ 4 + 3 and/or non-organ-confined disease) [32,60] and biochemical recurrence (BCR) after RP in men with clinically very low-, low- and intermediate-risk PC [60]. In addition, the use of GPS has been associated with the increased recommendation and utilization of AS in very low-, low- and favorable intermediate-risk patients because it can differentiate clinically indolent from aggressive PC and is designed to address heterogenicity, multifocality and limited biopsy sampling [32]. The 17-gene GPS was shown to be an independent predictor of AP in a prospectively designed validation study of a large, contemporary cohort of men with low- to low–intermediate-risk PC who were candidates for AS. Several publications have shown the role of Oncotype DX in predicting AP in male candidates for AS [60,61,62]. Cullen et al. [60] showed an association between GPS and the risk of recurrence in a racially diverse cohort who underwent RP. The analysis was conducted in biopsies of men with a very low, low and intermediate risk. GPS was a significant predictor of BCR in univariate analyses and after adjusting for clinical and pathology covariates. GPS was also a predictor of aggressive disease; however, the event number was too small and, therefore, inconclusive. In terms of metastatic disease, Van Den Eeden et al. [63] showed the association of Oncotype DX and the risk of metastatic disease in a 10-year risk study. They used diagnostic biopsy specimens of patients with low-, intermediate- and high-risk PC. The median follow-up was 9.8 years. Of the 259 patients evaluated, 79 had metastatic disease and 180 had non-metastatic disease. GPS was a significant predictor of time to metastasis, and GPS was independently associated with the National Comprehensive Cancer Network (NCCN) risk groups, American Urological Association (AUA) risk groups and Cancer of the Prostate Risk Assessment (CAPRA) score groups in multivariate analyses. In other analyses, GPS was also associated with BCR and prostate-cancer-specific death. In addition, Brooks and colleagues [64] studied the association of Oncotype DX and long-term outcomes (distant metastasis and prostate-cancer-specific mortality (PCSM) after RP) using tissue from the index lesions of RP specimens. GPS was independently associated with a 20-year risk of distant metastasis and PCSM. The studies by Van der Eeden et al. and Brooks et al. had similar results, even with independent cohorts. Van der Eeden et al. found that with a 20-unit increase in GPS, the hazard ratio for distant metastasis was 2.34 (95% CI, 1.42 to 3.86) and that for PCSM was 2.69 (95% CI, 1.50 to 4.82), and Brooks et al. showed equivalent results (HR, 2.24; 95% CI, 1.49 to 3.53 and HR, 2.30; 95% CI, 1.45 to 4.36, respectively). Interestingly, higher GPS scores were associated with more aggressive histological features, such as a cribriform pattern and a stromagenic pattern, as well as higher levels of progression in men in AS. The main limitation of these studies, as mentioned by the authors, is that they were exploratory, because the test was developed in the same sample. Therefore, external validation is necessary. To our knowledge, there are no further studies that associate Oncotype DX with the risk of metastatic disease at the time of diagnosis or after primary treatment. 

**Prolaris**. This commercial test is a gene expression classifier test that combines the CAPRA score with a cell cycle progression (CCP) score derived from a tumor RNA expression profile to produce a personalized metastasis risk score after definitive treatment for localized prostate cancer. The CCP molecular score was defined originally in 2011 in a retrospective cohort [65] and then was combined with CAPRA to create a Clinical Cell-Cycle Risk (CCR) score, first described in 2015 by Cusick et al. [28], using samples from needle biopsies in a cohort of patients on AS. The CCR score is a good predictor of outcomes in men who undergo primary treatment for localized PC in AS, surgery or radiotherapy [66,67]. Cusick et al. [28] published a validation study in a cohort of patients managed conservatively using CCP as a predictor of death. They used needle biopsies, and, in multivariate analyses, the CCP score’s overall hazard ratio was 1.76 (95% CI (1.44, 2.14)) and the CCR’s score was highly predictive, with a hazard ratio of 2.17 (95% CI (1.83, 2.57)). Swanson et al. published [68] a study of 360 men in 2021 using sample tissue from radical prostatectomy specimens, in which CCR was demonstrated to be an independent predictor of metastatic disease and disease-specific mortality after RP (HR = 3.03 [95% confidence interval (CI): 1.49, 6.20]; *p* = 0.003) and disease-specific mortality (HR = 3.40 [95% CI: 1.52, 7.59]; *p* = 0.004), respectively. Tward et al. addressed the potential benefits of using CCR to determine the need for adjuvant therapy in a study published in 2021 [69]. They used the CCR score and defined a multimodality therapy threshold for patients who had radiotherapy or surgery to assess the need for ADT or radiation, respectively. They studied patients with NCCN of unfavorable intermediate and high risk. They estimated the risk of progression in men who underwent surgery or RT using the prognostic value of CCR scores below and above the threshold, stratified by single- or multimodality therapy, for the prediction of metastasis. In all cases, men with CCR scores above the threshold who received single-modality therapy had the worst outcomes. The clinical utility of this study is that it can provide guidance about when to use adjuvant treatment in patients who undergo surgery or radiotherapy. For example, the authors found that the reduction in the Kaplan–Meier estimated risk for metastasis based on adding ADT to RT for the population of men below the threshold was 2.2%, whereas, for those above the threshold, it was 20.3%. In 2022, Tward et al. published an article [70] in which they addressed the utility of CCR scores in predicting the development of metastasis after primary radiation therapy. They also sought to validate the CCR score multimodality threshold described in their previous study in NCCN unfavorable intermediate-, high- and very high-risk patients, where RT alone, rather than the current recommendation of RT plus ADT, might be considered in these groups of patients. This was a multi-institutional study cohort of men with prostate cancer treated with dose-escalated external beam RT (EBRT) with or without ADT. The CCR score was highly prognostic for metastasis in the full cohort (HR, 2.22; 95% CI, 1.71–2.89; *p* < 0.001); however, the CAPRA score by itself was not a predictor of metastatic disease. In terms of the need for ADT associated with RT, the threshold was dichotomized as either below or above a CCR score of 2.112. The risk of metastasis below the threshold was low, regardless of the NCCN category. Furthermore, for men below the threshold, ADT of any duration did not significantly reduce the risk of 10-year metastasis as compared to patients treated only with RT (3,7% for each group). On the other hand, men in the above-threshold group who were treated with single-modality therapy had a more than six-fold greater predicted risk of developing metastasis compared to those below the threshold. 

**Decipher.** The Decipher^®^ gene signature consists of a 22-gene panel representing multiple biological pathways and was developed in 2013 [71] to predict systemic progression after definitive treatment. Originally, it was based on the expression of 22 RNA biomarkers related to androgen receptor signaling, cell proliferation, differentiation, motility and immune modulation using radical prostatectomy specimen tissue samples; however, currently, there is also a Decipher^®^ created from needle biopsy tissue samples. This has resulted in a final set of 22 markers corresponding to RNA from coding and non-protein coding regions of the genome and based on a majority rule criterion: the patients with a genomic classifier (GC) score greater than 0.5 are classified as high risk, whereas those with a score lower than or equal to 0.5 are classified as low risk. In multivariable analyses, after adjusting for post-RP treatment, GC remained the only significant prognostic variable (*p* < 0.001), with an odds ratio of 1.36 for every 10% increase in the GC score. The independent significance of GC suggests that a more direct measure of tumor biology (i.e., a 22-marker expression signature) adds significant prognostic information for the prediction of early metastasis after a rising PSA, which is not captured by the clinical variables available from pathological analyses. Cases with high GC scores died earlier from prostate cancer. In another study, a randomized clinical trial [72] using radical prostatectomy tissue specimens validated Decipher as an independent predictor of distant metastasis, with the secondary endpoints of prostate-cancer-specific mortality (PCSM) and overall survival (OS). Patients received salvage radiotherapy with or without 2 years of bicalutamide. In a multivariable analysis, GC (continuous variable, per 0.1 unit) was independently associated with distant metastasis (hazard ratio [HR], 1.17; 95% CI, 1.05–1.32; *p* = 0.006), PCSM (HR, 1.39; 95% CI, 1.20–1.63; *p* < 0.001) and OS (HR, 1.17; 95% CI, 1.06–1.29; *p* = 0.002) after adjusting for age, race/ethnicity, Gleason score, tumor stage, margin status, entry PSA and treatment arm. Interestingly, the estimated absolute effect of bicalutamide on 12-year OS was less when comparing patients with lower vs. higher GC scores (2.4% vs. 8.9%). Spratt et al. [73], in a meta-analysis of five studies including 855 high-risk patients with 8 years of follow-up, analyzed the performance of the Decipher^®^ genomic classifier (GC) test on men post-RP as a predictor of metastasis development. The 10-year cumulative incidence metastasis rates were 5.5%, 15.0% and 26.7% (*p* < 0.001) for patients classified by Decipher as low, intermediate and high risk, respectively. The authors showed in a multivariable analysis that Decipher was a statistically significant predictor of metastasis (HR: 1.30, 95% CI: 1.14–1.47, *p* < 0.001). In a systematic review published by Jairath et al. [74], which included 44 studies that addressed the role of Decipher in different settings, such as localized, postprostatectomy, nonmetastatic castration-resistant and metastatic hormone-sensitive PC, GC was independently prognostic for all study endpoints (adverse pathology, biochemical failure, metastasis, cancer-specific and overall survival) in multivariable analyses. They concluded that GC is most accurate for intermediate-risk PC and postprostatectomy decision making. 

In conclusion, Oncotype DX has more evidence and was designed for the detection of AP or aggressive histology, and the evidence for the prediction of distant metastasis is still poor because, in addition to the two studies described, there is a need for validation of the results and better-quality evidence. Prolaris (CCR) may have a role in determining the need for or avoidance of adjuvant therapy (either with RT or ADT) after RP or RT, but the literature and current evidence are not sufficient. Randomized clinical trials are necessary to demonstrate its role in indicating the need for adjuvant therapy and predicting metastatic disease. From our point of view, Decipher has the best evidence as a TBGB and can be used through RP specimens or needle biopsy specimens for the prediction of metastasis. Further studies are needed to establish how to best incorporate Decipher into clinical decision making. Currently, there are ongoing clinical trials utilizing the Decipher genomic classifier for PC. These trials are assessing the potential utility of Decipher in different stages, including its use in predicting the usefulness of different new antiandrogen therapies as primary and adjutant treatments. These clinical trials will be described later in the article. Appendix A provides a summary of the three TBGB described above. 

### 2.3. Impact on Clinical Guidelines and Clinical Practice

We performed a review of the present and current recommendations from different panels regarding the utility of different TBGB (Appendix A). In this review, we discuss the American Urological Association/American Society for Radiation Oncology (AUA/ASTRO), NCCN, European Association of Urology (EAU) guidelines and American Society of Clinical Oncology (ASCO) guidelines. Importantly, we summarize the recommendations at the time of this review, and, therefore, there could be some differences with future versions of the guidelines because they are dynamic and can be modified every few months or years. 

**AUA/ASTRO Guidelines 2022** [75,76,77]: (1)Clinically localized.prostate cancer guidelines. Guideline statement/risk assessment (part 1).“Clinicians may selectively use tissue-based genomic biomarkers when added risk stratification may alter clinical decision-making. (Expert Opinion)”. “Clinicians should not routinely use tissue-based genomic biomarkers for risk stratification or clinical decision-making. (Moderate Recommendation; Evidence Level: Grade B)”. In terms of the prediction of metastatic disease, the guidelines state clearly “two studies using biopsy data have shown that a cell cycle progression panel (Prolaris) score was associated with the risks of biochemical recurrence, metastatic disease, and prostate cancer death; however, only one of those studies met inclusion criteria for the systematic review”. The Oncotype Dx assay has been validated on needle biopsy tissue and found to be associated with adverse pathology, biochemical recurrence, metastasis and prostate cancer death; again, however, the studies did not meet the inclusion criteria for the systematic review. Meanwhile, a multi-institutional evaluation of Decipher biopsy testing found that a high-risk Decipher score was associated with conversion from active surveillance to definitive treatment. Thus, based on the level of existing data, the panel concluded that clinicians should not routinely use tissue-based genomic biomarkers for risk stratification or clinical decision making [75]. In part 3, Future Directions/Genomic Classifiers (GCs), the panel concludes, “The ability for commercially available GCs to improve the outcomes of patients with clinically localized prostate cancer has not been validated in prospective clinical trials to date. Prospective validation of the predictive capacity of GCs in localized disease will be important to support widespread use for treatment selection. Several ongoing clinical trials are indeed evaluating treatment intensification and de-intensification based on GC results in both intermediate- and high-risk patient populations”. (2)The advanced prostate cancer AUA/ASTRO/SUO guidelines, published in 2021 [78,79] and amended in 2023 [80], make no mention of genomic tissue biomarkers in the presence of biochemical recurrence without metastasis, MHSPC, nmCRPC or mCRPC in the statements of what clinicians should or may do for prognosis and treatment. In the 2021 guidelines part 2, in Future Directions, the organization states, “As we move forward as a field, we need to focus on the biologic make-up of tumors and how these can be better leveraged to identify treatment options for patients”. In the update published in 2023, in Future Directions/Biomarkers and Other Systemic Therapies, there is no mention of genomic tissue biomarkers, but rather of germline and somatic tumor alterations such as DNA damage response genes and DNA mismatch repair genes.

**NCCN guidelines. Version 1.2023** [81]. In Principles of risk stratification, the guidelines mention, “Patients with NCCN low, favorable intermediate, unfavorable intermediate, or high-risk disease and life expectancy ≥ 10 years may consider the use of the following tumor-based molecular assays: Decipher, Oncotype DX Prostate, and Prolaris” (PROS-D 2 of 4). Then, in a table (PROS-D 3 of 4), these three biomarkers are shown as prognostic but not predictive tools and only Decipher has an “endpoint trained for” or was designed to predict and optimize distant metastasis. In the same table, the level of evidence for the validation of Decipher is described as level 1, indicating “validation in the context of multiple clinical trials with consistent results. Randomized trials are necessary for predictive biomarkers for validation”. Prolaris is described as follows: “Like other biomarkers it has been validated for multiple endpoints, but the test was not specifically trained for an endpoint a priori”. Oncotype DX prostate is described as having a role in showing adverse pathology. Following the principles of risk stratification table, Prolaris and Oncotype DX are indicated to have level 3 validation, which means “Validation in multiple independent retrospective studies with consistent results”. None of these three biomarkers are part of the Initial risk stratification and staging workup for clinically localized disease (PROS-2). However, in a section titled PROS-2A underneath the table, there is an explanation: “Tumor-based molecular assays and germline genetic testing are other tools that can assist with risk stratification”. Decipher is not mentioned as one of the tools in a PSA persistence/recurrence patient (PROS-10). However, in the guideline’s development, in the section Tumor multigene molecular testing (MS-9), the panel recommends that the Decipher molecular assay should be used to inform adjuvant treatment if adverse features are found post-radical prostatectomy, The panel indicates that “patients with low or favorable intermediate disease and life expectancy greater than or equal to 10 years may consider the use of Decipher, Oncotype DX Prostate, or Prolaris during initial risk stratification. Patients with unfavorable intermediate- and high-risk disease and life expectancy greater than or equal to 10 years may consider the use of Decipher or Prolaris. In addition, Decipher may be considered to inform adjuvant treatment if adverse features are found after radical prostatectomy and during workup for radical prostatectomy PSA persistence or recurrence (category 2B for the latter setting). Future comparative effectiveness research may allow these tests and others like them to gain additional evidence regarding their utility for better risk stratification of patients with prostate cancer” (MS-10). 

**EAU guidelines:** 2023. [82,83] Chapter 6. 6.2.1.1.2. Tissue-based prognostic biomarker testing. Biomarkers, including Oncotype Dx^®^, Prolaris^®^, Decipher^®^, PORTOS and ProMark^®^, are promising. However, further data will be needed before such markers can be used in standard clinical practice. 6.2.5.2.1. Biomarker-based risk stratification after radical prostatectomy. The guidelines only mention Decipher. There is no mention of Prolaris or Oncotype. Decipher is described as follows: “The Decipher^®^ gene signature consists of a 22-gene panel representing multiple biological pathways and was developed to predict systemic progression after definitive treatment. A meta-analysis of five studies analyzed the performance of the Decipher^®^ Genomic Classifier (GC) test on men post-RP. The authors showed in multivariable analysis that Decipher^®^ GC remained a statistically significant predictor of metastasis (HR: 1.30, 95% CI: 1.14–1.47, *p* < 0.001) per 0.1 unit increase in score and concluded that it can independently improve prognostication of patients post-RP within nearly all clinicopathologic, demographic, and treatment subgroups. A systematic review of the evidence for the Decipher^®^ GC has confirmed the clinical utility of this test in post-RP decision-making. Further studies are needed to establish how to best incorporate Decipher^®^ GC in clinical decision-making.” However, there is no mention of recommendations for the use of Decipher or any other biomarker for the decision-making process in patients who undergo primary local treatment and then relapse. Furthermore, in chapter 6.3, Management of PSA-only recurrence after treatment with curative intent, there is no mention of a potential role of any biomarker. 

**ASCO guidelines.** For localized prostate cancer, published in 2018 [84], there is only a mention in point 32 of Appendix A regarding AS. “Tissue based genomic biomarkers have not shown a clear role in active surveillance for localized prostate cancer and are not necessary for follow up. (Expert Opinion).” There is no mention of tissue biomarkers in high-risk patients or post-prostatectomy or radiotherapy follow-up for the prediction of metastasis or their role in the decision making for subsequent therapy. Then, in the section on molecular biomarkers in localized prostate cancer, published in 2020 [85], the guidelines delineate four questions (Appendix A of the guidelines). The first one is related to the role of biomarkers in the selection of patients for AS, the second one is concerned with the usefulness of biomarkers for the diagnosis of clinically significant prostate cancer, the third is related to the role of biomarkers in the decision making on adjuvant or salvage therapy after radical prostatectomy, and the fourth is concerned with the comparison of genomics vs. MRI in identifying clinically significant prostate cancer. In response to questions 1 and 2, the panel’s recommendation is that a “routine ordering of molecular biomarkers is not recommended” based on insufficient evidence with a moderate grade of recommendation. In their answer to question 3, they mention the Decipher genomic classifier as the only option; however, “in the absence of prospective clinical trial data, routine use of genomic biomarkers in the post prostatectomy setting to determine adjuvant versus salvage radiation or to initiate systemic therapies should not be offered”, based on insufficient evidence. Finally, for the fourth question, the panel recommends using them only when the results are likely to affect clinical management. These tests may provide information independent of the clinical parameters and independent of one another, indicating a weak grade of recommendation. 

Finally, in a section on non-castrated advanced, recurrent and mPC, updated in 2023 [86], there is no mention of biomarkers in predicting metastasis or their usefulness in the management of these patients. 

In conclusion, the panels of the four associations that we analyzed are very consistent in that TBGB may be utilized in some cases; however, there is not enough evidence to support a recommendation that tissue biomarkers should be used. Nevertheless, biochemical, histological and clinical parameters have stronger evidence and, therefore, the guidelines place them above tissue biomarkers as a tool for biochemical recurrence and metastatic development. Our belief is that positive or negative biomarkers are a very important tool to improve the risk probability of recurrence and eventually of adjuvant treatment or closing follow-up. 

### 2.4. Current Prostate Cancer Treatments and Potential New Therapeutic Opportunities Using Tissue-Based Biomarkers for Prediction of Metastasis

There is a great need for biomarkers to determine which patients will develop metastasis in the future, to better determine the primary treatment. We describe current treatments according to the evidence for different stages, where a TBGB tissue-based biomarker could play an important role in modifying and improving treatment. 

**Localized prostate cancer.** For localized low- and favorable intermediate-risk PC, currently, the best treatment options are radical prostatectomy, radiation therapy or AS. Biomarkers to define the best candidates for these treatments in AS are not addressed in the current review because they were reviewed in a previous work [19]. For unfavorable intermediate- and high-risk PC, current treatments include RP plus extended lymphadenectomy, radiotherapy plus ADT and possibly abiraterone for high-risk patients [76,82,84]. PSMA PET/CT has a relatively new role in staging high-risk prostate cancer patients. According to a proPSMA multicenter randomized phase 3 study published by Hofman et al. in 2020 [21], PSA had 27% greater accuracy than conventional imaging in detecting metastasis as a first-line imaging modality and also showed a higher rate of management changes than conventional imaging (28% vs. 15%). For TBGB, currently, there is no clear role in staging or defining the presence of micrometastasis or predicting clinical metastasis, as explained above. The evidence is still limited and based on non-randomized clinical trials. There is a need for TBGB to better define, for example, which patients will require adjuvant treatment after local primary treatment. These biomarkers should not compete with imaging, but they should complement each other.

**Biochemical recurrence:** Defined as two consecutive PSA values of ≥0.2 ng/mL post-RP or 2 points above nadir post-radiation therapy, biochemical recurrence is an attractive subject, and the discussion is still ongoing regarding how to manage these patients. Histology variables such as the Gleason grade, tumor volume/extension and margin status are predictors of relapse after primary treatment. In the worst-case scenario, patients with ISUP grade > 2 in combination with EPE (pT3a) and particularly those with SV invasion (pT3b) and/or positive surgical margins are at a high risk of progression, which can be as high as 50% after 5 years [87]. LN involvement, capsular penetration of LN, LN density and the number of LN involved are also good predictors of recurrence [24,88,89]. PET CT has been demonstrated to detect recurrence post-primary treatment at a rate of 59–66%, localization at a rate of 84–87% and a change in management in 64% of patients at this stage [23]. Currently, treatment for biochemical recurrence is mostly based on radiotherapy and sometimes ADT. There is a major potential role for a TBGM in this setting in terms of predicting biochemical recurrence and defining which patients with BCR are better candidates for a new type of treatment. For example, if a patient is ISUP group 2 or 3, with a negative margin, but the biomarker is highly predictive of the development of metastasis, such a patient may be a good candidate for early radiotherapy or systemic treatment. 

**Adjuvant (ART) vs. salvage radiotherapy (SRT).** ART vs. early SRT and the efficacy of adjuvant ADT have been compared in three prospective randomized clinical trials (RCTs): the Trans-Tasman Oncology Group (TROG) Radiotherapy Adjuvant Versus Early Salvage (RAVES) trial [33], the Medical Research Council (MRC) Radiotherapy and Androgen Deprivation In Combination After Local Surgery (RADICALS) trial [34] and the Groupe d’Etude des Tumeurs UroGenitales (GETUG-AFU 17) [35]. A meta-analysis including all three of them has been published [36]. The three RCTs showed no differences between the two treatments in terms of biochemical progression-free survival. The studies showed that SRT had lower levels of ≥2 grade complications. Additionally, in recent years, ultrasensitive PSA has been introduced. Taking into account both of these factors in favor of SRT, the use of ART has shifted to early SRT.

**Clinically positive lymph nodes (N1).** Patients with clinical N+ disease require a multimodal therapy based mainly on radiotherapy plus ADT ± abiraterone or radical prostatectomy plus extended lymphadenectomy in selected cases [76,82]. There is no literature on the use of tissue-based biomarkers in patients with clinical N+. Biomarkers may be useful in this scenario, in which multimodal therapy is used in practically all patients. 

Today, for patients with HSPC and mCRPC that are already advanced, tissue-based biomarkers for the prediction of metastasis are not necessary because they already have metastasis; however, in the group of nmCRPC patients, the disease can be reduced substantially due to the use of PET PSMA, and because fewer patients are being treated with antiandrogen therapy in the context of biochemical recurrence. This group of patients does not develop metastasis. Well-known randomized trials that showed that apalutamide, enzalutamide and darolutamide improved metastatic, disease-free survival by 2 years in men who had a PSADT < 10 months. For men with PSADT > 10 months, observation is the best management option [47,90,91]. 

## 3. Conclusions and Future Directions

This review was motivated by the need for better-known tools that for the prediction of PC metastasis. We performed a review of the literature in regard to the importance of diagnosing mPC, the TBGB that currently exist that can play a role in this prediction, the recommendations of different urological and oncologic guidelines and finally recommendations for current treatment and the potential space for new therapies based on mPC prediction through TBGB. The importance of predicting metastasis is fundamental, given that, once metastasis is diagnosed, QoL and survival drop dramatically. Oncotype DX, Prolaris and Decipher are current TBGB that are on the market. Prolaris and Oncotype DX were designed primarily to predict local aggressiveness rather than distant metastasis. Decipher was designed as a predictor of metastasis using radical prostatectomy tissue specimens, and, more recently, using needle biopsy specimens. Today, it is the only TBGB tool that can predict metastatic disease and it has been used mainly with unfavorable intermediate- and high-risk patients to define better which patients should receive ADT complementing RT, and in biochemical relapse scenarios to define the best timing to initiate RT. However, there is still no strong evidence of its role in the management of unfavorable intermediate- and high-risk non-mPC. We understand that, currently, there are two active phase III trials evaluating the intensification or de-intensification of hormonal treatment according to risk-based genomic tissue biomarkers. In these trials, the main outcome is metastasis-free survival. The first one, the PREDICT-RT Trial (ClinicalTrials.gov ID NCT04513717), uses Decipher as a tissue biomarker and compares less intense hormone therapy and radiation therapy with traditional hormone therapy and radiation therapy in treating patients with high-risk (NCCN) prostate cancer and low gene risk scores. This trial also compares more intense hormone therapy (apalutamide) and radiation therapy to traditional hormone therapy and radiation therapy in patients with high-risk prostate cancer and high gene risk scores. In the second one, the Guidance Trial (ClinicalTrials.gov ID NCT05050084), patients with unfavorable intermediate prostate cancer (NCCN) and higher Decipher risk scores are assigned either to the use of 6 months of the usual treatment (hormone therapy and radiation treatment) or to the use of darolutamide plus the usual treatment (intensification). On the other hand, patients with low Decipher risk scores are assigned to the part of the study that compares the use of radiation treatment alone (de-intensification) to the usual approach (6 months of hormone therapy plus radiation). The final results of these trials will be important to better define whether patients with non-metastatic unfavorable intermediate and high risk, using the only predictor of metastatic disease available on the market (Decipher), will benefit from the intensification or de-intensification of systemic treatment and address its definitive role and usefulness in prostate cancer management. However, additional clinical trials will be necessary to answer this question. Guidelines such as AUA/ASTRO, NCCN, EAU and ASCO do not assign a specific role to tissue-based biomarkers, especially in predicting metastatic disease, due to a lack of strong evidence and consequently their underutilization. Current treatments for non-mPC cases, including localized cases, patients with BCR and clinical N+, are based on strong evidence for localized and weaker for BCR and clinical N+. With the high rate of biochemical relapse in unfavorable intermediate- and high-risk PC, the best predictors are clinical, biochemical and histological. Treatment and adjuvant therapies have not shown better outcomes over observation, probably because the tools used to define candidates for adjutancy are of low quality. 

The use of TBGB for metastasis in PC aids the clinician’s therapeutic decision making, focalizing the treatment according to the cancer’s aggressiveness. Although it is fundamental that a biomarker be of use for clinical practice, these biomarkers must be accessible to patients, but they currently cost USD 3000–5000. They also must be easy to implement from the point of view of the patient and the physician, with the aim of scaling quickly, becoming universal and benefiting all patients with newly diagnosed prostate cancer. Therefore, we believe that there is still a need and space for more cost-effective TBGB tests that predict mPC disease.

## Data Availability

Data sharing not applicable.

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
