# Peer review of "Can We Predict Prostate Cancer Metastasis Based on Biomarkers? Where Are We Now?"

_ijms, 2023, doi:10.3390/ijms241512508_

Round 1

Reviewer 1 Report

Dear authors of “Can we predict prostate cancer clinical metastasis based on biomarkers? Where are we now?”,

Thanks for giving me the opportunity of reviewing your manuscript. This is an interesting state of art on the use of tissue-based genomic biomarkers (TBGB) for predicting metastatic disease. The manuscript is well written and organized. It is included updated on recent articles. Such a matter is a great matter and still a point of high Research over the planet. All the aspects are almost cover. Nevertheless, there are still minor points which need to be solved before its acceptance for publication.

Surprisingly, there are no figures or tables within the manuscript which is quite peculiar for a review in IJMS. Are the two mentioned tables in the text the ones provided as supplementary data?

The introduction starts with a sentence focusing on PC situation in US whereas the review seems to encompass worldwide data. On suggestion is to move or delete this first paragraph.

Also, with the sentence: “We can obtain these biomarkers through needle biopsies or RP tissue specimens - years before the theorical appearance of metastasis - in order to cure patients before they have metastasis but with a high risk of developing it or even with circulating tumor cells (CTC) waiting to nest in the bones and/or lymph nodes”, it seems that such biomarkers already exist whereas there are still research to identify and use them. Please correct.

In the paragraph introducing the following part of the manuscript, and to my point of view, it is not compulsory to detail the mane and company of the available commercial kits. “Which tests are available to predict metastatic disease? What is the evidence and the efficacy of commercialized tissue prognostic tests Prolaris (Myriad Genetics), Oncotype DX Prostate (Exact Sciences), and Decipher (Genome DX Biosciences)?” It is like already giving the answer without having to read this part of the manuscript.

This second section is less easy to read in the manuscript. There are many used abbreviations which are not even defined: NCCN, AUA, CAPRA, CCR, GC, CC, GCC, etc. Please define them and try to rewrite by making this section easier to be read. In the same vein, there are no conclusions at the end of this section. Some remarks or discussion on their capacity and efficiency to detect and discriminate mPC are welcome, maybe by precising that such points will be addressed regarding the recommendations of the Scientific Societies over the world.

Text formatting should be homogenous within all manuscript. In page 4, one subject is missing “Therefore, is crucial to treat patients during localized disease and eventually before detecting any signs of clinical metastasis.” and third paragraph should start by mPC and not MPC.

Looking forward seeing your modifications,

All the best,

Author Response

Dear reviewer,

We attached our responses to your comments. The tables are in a different document directly to the journal 

Thank you for your work

Reviewer 2 Report

This review article discusses the possibilities to predict prostate cancer clinical metastasis based on biomarkers.

General comment:

The article is well written and the topic is highly relevant. The information is presented in concise way, yet the article is very informative.

Specific point:

Why PORTOS signature is not described in more detail? The PORTOS study (DOI: 10.1016/S1470-2045(16)30491-0) says 'The primary endpoint was the development of distant metastasis'.

Minor points:

In Introduction, the abbreviation PSMA should be explained.

In some instances, 'Oncotype' is not written with capital letter.

Minor editing of English language required

Author Response

Dear reviewer,

We attached in a word document our responses to your comments,

Thank you for your work
